# Variations of Internal and External Load Variables between Intermittent Small-Sided Soccer Game Training Regimens

**DOI:** 10.3390/ijerph16162923

**Published:** 2019-08-15

**Authors:** Filipe Manuel Clemente, Pantelis Theodoros Nikolaidis, Thomas Rosemann, Beat Knechtle

**Affiliations:** 1School of Sport and Leisure, Polytechnic Institute of Viana do Castelo, 4960-320 Melgaço, Portugal; 2Instituto de Telecomunicações, Delegação da Covilhã, 6200-001 Covilha, Portugal; 3Exercise Physiology Laboratory, 18450 Nikaia, Greece; 4Institute of Primary Care, University of Zurich, 8091 Zurich, Switzerland; 5Medbase St. Gallen Am Vadianplatz, 9001 St. Gallen, Switzerland

**Keywords:** association football, drill-based tasks, intermittent exercises, physiological, physical, performance

## Abstract

The purpose of this study was twofold: (i) analyze the variations of internal and external load between intermittent regimens (6 × 3’ and 3 × 6’) during a small-sided game (SSG); and (ii) analyze the variations of internal and external load within-intermittent regimens (between sets). Ten male amateur soccer players (age: 21.7 ± 2.1 years) participated in this study. Almost certain large decreases in total distance (−8.6%, [−12.3; −4.8], Effect Size (ES): −1.51, [−2.20; −0.82]) and running distance (−34.0%, [47.0; −17.8], ES: −2.23, [−3.40; −1.05]) were observed when comparing the 3 × 6’ and 6 × 3’. Very likely moderate and large decreases in total accelerations (−24.0%, [−35.1; −10.9]; ES: −1.11, [−1.75; −0.47]) and total of decelerations (−26.7%, [−38.8; −12.1]; ES:−1.49, [−2.36; −0.62]), respectively, were found when comparing the 3 × 6’ and 6 × 3’. Very likely increases in rated of perceived exertion in the set 3 in comparison to the 1st during the 3 × 6’ SSG (34.5%, [12.4; 61.0], ES: 1.35, [0.53; 2.16]) and the 6 × 3’ (29.9%, [11.6; 51.2]; ES: 1.17, [0.49; 1.85]). Longer sets increase the perception of effort and contribute to a large decrease in total and running distances, and total of accelerations and decelerations. Meaningful decreases in time-motion demands occur between sets 2 and 3 while perceived effort increases.

## 1. Introduction

Small-sided games (SSGs) are very popular exercise drills designed by coaches to replicate official match dynamics and increase the intensity and individual participation of players during soccer training sessions [1,2]. In SSGs, the format of play (number of players involved), pitch size, and some rules can be manipulated to adjust the exertion required by players to meet the coach’s proposed objective [3,4]. One of the advantages of SSGs is that, if properly designed, they may represent an effective strategy for multicomponent training [5], allowing for the development of both physical/physiological and technical/tactical skills at the same time [6].

SSGs are often used to promote new affordances and to adjust the tactical complexity to the main goal of the coach, improving the decision making of players [7]. In fact, small variations in these games may promote a significant change in the player’s behavior, thus resulting in consequences for the overall intensity of exercise [8]. Despite these games being promoted for improving the collective and individual behavior of players, there is also a relationship with the physical and physiological demands of players.

Among the common topics researched in SSGs and its physiological effects is the effects of training regimens on players’ performance and acute responses [9]. The load imposed on players and the way in which this load occurs must be understood to ensure that training stimuli are adjusted to promote high-intensity exertion without compromising external load or technical/tactical performance [10]. The load is closely related to the following training prescription that should be taken in account [2]: (a) work intensity and duration; (b) recovery type (rest/active recovery) and duration; and (c) total work duration (work interval number × work duration).

In the specific case of SSGs, comparisons between continuous and intermittent training regimens are commonly conducted [9,11]. Most results do not reveal meaningful or clear changes between these types of regimens in terms of heart rate responses and blood lactate concentrations [9,12]. However, when comparing different intermittent regimens (short, moderate, and long) with continuous regimens, the evidence suggests that continuous regimens result in higher values of maximal heart rate, blood lactate concentration, and perceived exertion [13]. Moreover, continuous regimens seem to increase the distance covered at low running speeds and decrease moderate-to-intense running distances [13].

As high-intensity drills, SSGs seem to fit in the category of interval training. Rather than compare the acute effects of continuous versus intermittent regimens, it is important to analyze the effects of different intermittent regimens. In a study conducted using a 3 × 3 SSG format, it was found that long bouts (sets) (3 × 6 min/2 min rest) decreased heart rate responses in comparison to medium (3 × 4 min/2 min rest) and small (3 × 2 min/2 min rest) bouts [14] if the first minute of data is excluded. However, no meaningful changes were observed in perceived exertion or technical actions [14]. In another study, it was found that, compared to long bouts, shorter bouts elicited lower maximal heart rate, shorter total distance covered at low running speed, and greater distances covered at medium and high running speeds [13].

The proper adjustment of SSGs to suit the purpose of training may help coaches optimize the amount of exertion imposed on players and may improve their performance. However, while a couple of experiments have compared different intermittent soccer training regimens [13,14], the information presented lacks a clear demonstration of the effects of different training regimens on internal (acute physiological responses) and external (physical demands) load variables. A comparison between bout durations should help researchers understand the patterns of exertion and provide meaningful information to coaches to help them choose the most effective regimens for their players. Moreover, coaches will also be able to analyze the effects of different intermittent regimens on time-motion performance during SSGs. Based on this rationale, the aim of the present study was two-fold: (i) to analyze variations in rate of perceived exertion, heart rate responses, and time-motion demands between two intermittent training regimens (6 × 3’ and 3 × 6’) during an SSG (5 × 5 format); and (ii) to test the variations of the above-mentioned variables within training regimens (between sets).

## 2. Materials and Methods

### 2.1. Participants

Ten male amateur soccer players (age: 23.7 ± 1.1 years; experience: 10.3 ± 3.1 years; height: 178.2 ± 5.3 cm; weight: 72.1 ± 4.9 kg) competing at the regional level participated in this study. The participants usually trained three times a week and played one match every week. Participants were informed about the study design and the potential implications, risks, and benefits of participating. After that, participants freely signed an informed consent. The experiment followed the ethical standards of the Declaration of Helsinki for the study in humans. The study was approved by the local ethical committee (School of Sport and Leisure) with the code number IPVC-ESDL180503.

### 2.2. Experimental Approach

This study used a counterbalanced repeated-measures design to compare the rate of perceived exertion (RPE), mean heart rate (HRmean), total distance (TD), running distance (RD), sprinting distance (SD), total accelerations (TAc), total decelerations (TDc) and player load (PL) of the participants in two different SSG regimens: 6 sets of 3 min with 2 min of rest (6 × 3’ regimen) and 3 sets of 6 min with 2 min of rest (3 × 6’ regimen). The study was conducted for 2 weeks immediately after the last official match of the season. The study occurred in the same pitch and the 10 players participated in all data collection sessions.

The 5 × 5 format was employed in both training regimens. Each regimen was implemented twice, interspaced by a period of one week to ensure each session was performed under similar conditions. In each session, only one SSG regimen was implemented. In the first week, the 6 × 3’ regimen was implemented first and the 3 × 6’ regimen was implemented 48 h afterward, without any other training sessions in between. In the second week, the inverse sequence was employed. The mean of results obtained in each condition (mean of the two days of data collection with the same regimen) was used between regimens comparisons.

The players were distributed into two teams based on skill level and playing position to homogenize the competitive level. The teams did not change during the study. Players wore vests equipped with a GPS and HR sensors during the SSGs. The SSGs were played on synthetic turf at 6:00 p.m. at an average temperature of 23 °C and a relative humidity of 57% no SSGs played under rainy weather conditions. All SSGs were preceded by a standardized warm-up consisting of 5 min of jogging, 5 min of lower-limb dynamic stretching and mobility exercises, 5 min of agility and speed drills, and 5 min of a ball possession game.

### 2.3. Small-Sided Game

The 5 × 5 format was implemented with small goals (2 × 1 m) without goalkeepers. The size of the pitch was 42 × 22 m (924 m^2^). The individual playing area (area divided by the total number of players) followed previous recommendations to promote SSGs based on the real game situations (area of play) in attacking processes [15]. Two training regimens were used: 6 × 3’/2’ rest and 3 × 6’/2’ rest. The teams were composed of 2 defenders, 2 midfielders and 1 forward with similar skill levels as based on a preliminary observational test.

No specific verbal instructions were provided before, during, or after the SSGs. However, verbal encouragement was provided to keep players committed and to help them to maintain a high exertion level. Six balls were placed around the pitch to ensure a quick repositioning if the ball in play went out of bounds. The SSGs followed official soccer rules with exception of offside.

### 2.4. Rating of Perceived Exertion (RPE)

Players were instructed to rate their perceived effort immediately after each SSG. The CR-10 point scale [16] was used to classify the effort; on this scale, 1 means “very light activity” and 10 means “maximal exertion.” Players rated the effort individually as to not hear or be influenced by other teammates’ responses. All the players were previously instructed of the use of the scale to optimize the accuracy of their ratings during the experiments.

### 2.5. Heart Rate (HR) and Global Positioning System (GPS)

Players wore an HR sensor and a chest belt (Polar H7, Polar Electro, OY, Kempele, Finland) which recorded data every second during the SSGs. Data were imported into the Polar Team application. The HRmean (bpm) per each set was used to measure this variable.

The players also wore a vest with a geolocation tracker (JOHAN Sports, Noordwijk, The Netherlands) consisting of a GPS sensor (10 Hz, including EGNOS correction), accelerometer, gyroscope, and magnetometer (100 Hz, 3 axes). The validity and reliability values of the devices can be found in a previous study [17]. The tracker was placed in a bag of the vest located at the dorsal region. The data was exported and treated immediately after each session.

The following variables were collected from the tracker devices: (a) total distance (meters per minute); (b) running distance at 14–19.9 km/h (meters per minute); (c) sprinting distance at >20.0 km/h (meters per minute); (d) total accelerations >2 m/s^2^ (number per minute); (e) total decelerations >2 m/s^2^ (number per minute); and (f) player’s load (g per min), being calculated by estimating the total acceleration difference between two consecutive time steps being the length of the three-dimensional vector of accelerations in the anteroposterior, mediolateral, and craniocaudal axes between time step = 0 and time step = 1. Players were familiarized with the use of the HR belts and trackers vests before the study began.

### 2.6. Statistical Procedures

The results are presented as either means and standard deviations (SD) or percentage differences and 90% confidence intervals (90% CI). The confidence intervals were defined following the recommendations for this kind of sample [18]. Normality and homogeneity of the data were firstly tested and verified before the inference analyses. Between-training regimens and within-training regimens differences were analyzed using the standardized differences of the effect size (ES) [19], with a 90% CI. ES was classified as trivial (<0.2), small (0.2–0.6), moderate (0.6–1.2), or large (>1.2) [18]. Probabilities were calculated considering the smallest worthwhile changes (SWC, 0.2 × between-subjects SD) [20]. Qualitative probabilistic mechanistic inferences of the true effects were made using these probabilities [20]. The scale for qualitative probabilities was as follows: 25–75% = possible; 75–95% = likely; 95–99% = very likely; >99% = almost certain [20].

## 3. Results

Descriptive statistics of internal and external load variables in both SSG training regimens can be found in Table 1 (values represent the average of the two sessions per type of intermittent protocol). Descriptive analyses reveal that RPE were higher in the last set in both 6 × 3’ (6.1 ± 1.9 arbitrary units (A.U.)) and 3 × 6’ (6.7 ± 1.6 A.U.) training regimens. The HRmean was higher in the second set of 6 × 3’ regimen (171.6 ± 10.0 bpm) and in the third set of 3 × 6’ regimen (171.1 ± 10.9). TD was greater in the third set of 6 × 3’ regimen (112.5 ± 11.1 m/min) and in the first set of 3 × 6’ regimen (103.3 ± 7.6 m/min). RD was greater in the second set of 6 × 3’ regimen (10.0 ± 5.1 m/min) and in the second set of 3 × 6’regimen (9.4 ± 5.6 m/min). SD was greater in the third set of 6 × 3’ regimen (1.5 ± 1.9 m/min) and in the second set of 3 × 6’ regimen (0.7 ± 1.3 m/min). TAc and TDc were greater in the second set of 6 × 3’ regimen (2.9 ± 0.8 and 2.7 ± 0.9 n/min, respectively) and in the first set of 3 × 6’ regimen (2.1 ± 1.0 and 1.9 ± 1.0 n/min, respectively). Finally, PL was greater in the second and third sets of 6 × 3’ regimen (7.3 ± 1.3 g/min) and in the first set of 3 × 6’ regimen (1.9 ± 1.0 g/min).

Between-training regimen variations can be found in Table 2 (values represent the average of the two sessions per type of intermittent protocol). Almost certain large decreases of TD (−8.6%, [−12.3; −4.8], ES: −1.51, [−2.20; −0.82]) and RD (−34.0%, [47.0; −17.8], ES: −2.23, [−3.40; −1.05]) were observed when comparing 3 × 6’ versus 6 × 3’ regimens. Very likely moderate and large decreases of TAc (−24.0%, [−35.1; −10.9]; ES: −1.11, [−1.75; −0.47]) and TDc (−26.7%, [−38.8; −12.1]; ES: −1.49, [−2.36; −0.62]), respectively, were found when comparing 3 × 6’ versus 6 × 3’ regimens.

Within-training regimen differences can be found in Figure 1 and Figure 2. To simplify the analysis, the six sets within 6 × 3’ SSG regimen were grouped in three sets consisting the first set in the average of sets 1 and 2, the second set in the average of sets 3 and 4 and the third set in the average of 5 and 6. Very likely increases of RPE were found in the 3rd set in comparison to the 1st during the 3 × 6’ SSG regimen (34.5%, [12.4; 61.0], ES: 1.35, [0.53; 2.16], large magnitude) and 6 × 3’ SSG regimen (29.9%, [11.6; 51.2]; ES: 1.17, [0.49; 1.85], moderate magnitude). Trivial-to-small changes of HR were found between sets in both training regimens. Likely decreases of total distance were found from set 3 to set 1 (−11.6%, [−20.8; −1.3], ES: −1.78, [−3.38; −0.19], large magnitude) and from set 2 to set 3 (−9.2%, [−18.6; 1.3], ES: −1.33, [−2.85; 0.18], large magnitude) in 3 × 6’ SSG regimen. Very likely decreases of total distance were found from set 3 to set 1 (−6.2%, [−8.8; −3.5], ES: −0.96, [−1.39; −0.53], moderate magnitude) and almost certain decreases were found from set 3 to set 2 (−6.0%, [−8.0; −3.9], ES:−0.87, [−1.18; −0.56], moderate magnitude) in 6 × 3’ SSG regimen. Very likely decreases of player load were found from set 3 to set 1 (−13.1%, [−23.1; −2.9], ES: −1.37, [−2.46; −0.27], large magnitude] during 3 × 6’ SSG regimen.

Within-training regimens differences in running distance were trivial-to-small in the majority of comparisons and the moderate magnitudes were unclear. Considering the sprinting distances, the variations were also trivial-to-small. Unclear large (−30.6%, [−58.3; 15.5], ES: −1.96, [−4.69; 0.77], *large magnitude*) and unclear moderate (−24.9%, [−52.3; 18.7], ES: −1.18, [−3.08; 0.71], *moderate magnitude*) decreases of total accelerations were found from set 3 to set 1 and set 3 to set 2 during 6 × 3’ SSG regimen, respectively. Trivial-to-small changes of total decelerations and unclear moderate differences were found in both training regimens.

## 4. Discussion

Between-SSG training regimens, changes were observed in the present study. Almost certain large increases in total and running distances were observed in the shorter sets (6 × 3’), and very likely moderate and large increases in total accelerations and decelerations, respectively, during shorter sets were found. RPE showed likely small increases during longer sets (6 × 3’). Likely small increases in player load during shorter sets were also found. Briefly, this evidence suggests that shorter sets contribute to an increase in terms of moderate-running speed distances and high-intensity actions associated with accelerations and decelerations higher than 2 m/s^2^ while resulting in a lower RPE than in longer sets. Although no meaningful differences were found in a study that compared time-motion variables between 4 × 4’ and 2 × 8’ 5 × 5 SSG regimens [11], our results are partially in line with the findings of a study that compared short (6 × 2’), medium (3 × 4’), and long (2 × 6’) regimens [13].

Heart rate responses were trivially different between regimens, suggesting that this variable is not sensitive to variations in the intermittent regimens tested in our study. These results are in line with previous studies that compared heart rate responses after different intermittent regimens [11,14]. However, the alternative internal load marker used in our study (RPE) revealed a likely small increase during longer sets, suggesting that raising RPE may compromise the consistency of physical demands across sets.

Comparisons within SSG regimens were also performed to analyze the variations between sets. Main variations were found in RPE, total and running distances, and player load. Intriguingly, higher-intensity physical demands (sprinting distance and total accelerations and decelerations) were relatively constant across sets.

Moderate and progressive increases in RPE were found across sets, mainly during the 3 × 6’ regimen. The progressive increase in RPE scores was also found during shorter bouts. Trivial-to-small differences were found between the third and second sets. Eventually, longer sets may contribute to a perceptual increase of effort [21]. However, the fatigue effect may be the cause of such RPE increases, especially considering that decreases in total and running distances and player load occurred across the sets in both regimens.

Despite these progressive decreases, it was observed that shorter sets are probably more beneficial to delaying the impact of fatigue on total distance because large decreases were found between sets 1 and 3 and between sets 2 and 3 during the 3 × 6’ regimens, and only moderate effects were found during the 6 × 3’ regimens. Similar evidence was found regarding running distance. Moderate decreases were found across the sets during longer sets, and only small differences were observed during shorter sets. In both regimens and for both variables (total and running distance) trivial-to-small decreases were found from set 1 to set 2, suggesting that the main effects of fatigue emerge from the continuity of exertion [22].

The greater constancy of results in terms of sprinting distance and total of accelerations and decelerations across the sets may suggest that the resting period of 2 min was enough to maintain intensity levels and to maximize energy phosphates as the primary energy source [23]. However, in the specific case of sprinting distance, the values per minute did not reach 1 m on average; thus, the constancy can also be justified by the low frequency of these demands across the 5 × 5 format.

Interestingly, knowledge about the period of time may also constrain the pace of players during the games. In fact, a previous study that compared different intermittent regimens and their effects on pacing revealed that high-speed distances progressively and largely decreased across shorter sets (1 min) based on the ‘all-out’ pacing strategy used by rugby players [22]. Conversely, during longer sets, a more constant high-speed pace across the sets was observed [22]. The results of the present are in line with these previous findings. However, this constancy in the most intense activities that occurs in longer sets also resulted in smaller values when compared with shorter sets.

Our study had some limitations. The number and the competitive level of the players may constrain the inferences of this study. Theoretically, professional players present greater aerobic and anaerobic capacities, thus making it possible to reduce the magnitude of decreases throughout the bouts. However, the main results are in line with those of previous studies conducted in elite youth players [13], amateurs [11], and professionals [14]. In addition, our study analyzed the effects in only one SSG format, and for that reason, the occurrences of sprinting distances are scarce. Bigger formats should be considered for testing variations between regimens. Different work-to-rest ratios should be considered in future study designs in order to analyze the effects of rest on physical demands and the internal effect of the exercise. In fact, in our study the difference between training regimen also constrained the time of recovery and the work-to-rest ratios were different between regimens, thus this may interfere with some generalizations that can be made from our study. The small number of participants should also be considered as a limitation for a possible generalization of the evidence. Other conditions should also be understood as contextual and for that reason may affect comparisons with future studies, namely: (i) the synthetic grass was not wet; (ii) the same study conducted in another period of the season may conduct to different evidence; (iii) possibly more sessions would lead to a better copy with the decrements throughout the bouts; and (iv) different task conditions that affect the player’s behavior and, consequently, the intensity of exercise [24].

Despite these limitations, this study revealed that different training regimens led to different effects on internal and external load variables. The main highlights of the present study are that shorter sets (3’) seem to be more beneficial than longer sets (6’) in keeping total and running distances and total accelerations and decelerations constant while decreasing the RPE. Additionally, the greater decrements seem to mostly occur in the last sets. Despite the somewhat predictable results, the small amount of previous research into such specific issues highlights the innovative character of the present study to reveal that shorter sets contribute to a greater external load stimulus despite that no meaningful changes were found between training regimens in the internal load. This may suggest that the internal load measures cannot be the unique criteria to choose between bout periods. In fact, in the present study it is possible to observe that despite the similarity of internal load between intermittent regimens, meaningful increases of some important external load measures were found (e.g., total distance, running distance, accelerations and decelerations).

As practical implications, we may hypothesize that smaller periods of exertion with a greater number of sets (bouts) can be recommended to ensure a more constant high level of physical demand and to contribute to an optimization of the high-energy systems that support highly demanding actions. Possibly, longer bouts should be used in situations in which coaches want to develop aerobic capacity while players experience acute fatigue effects and consequent decrements in the external load. Moreover, shorter sets enhance a meaningful greater stimulus in terms of distance covered, running distance and total number of accelerations and decelerations. However, for a better adjustment of the training regimen with the reality of the match, it will possibly be interesting in the future to compare the regular periods of high-intensity effort in the match and adjust such periods in SSGs, thus making more real the period of high-exertion and the work-to-rest ratios.

## 5. Conclusions

Between-SSG training regimens revealed that shorter sets (6 × 3’) almost certainly largely increased total and running distances and very likely moderately and largely increased total accelerations and decelerations, respectively, in comparison to longer sets (3 × 6’) while likely small increases in RPE were found in longer sets. The within-regimen analyses revealed that longer sets contributed to increases in RPE across sets and to large and progressive decreases in total distance and player load across sets. Additionally, moderate decreases in running distance and total decelerations were found progressively across shorter sets, while these variables were more stable between longer sets. The overall conclusions should be faced carefully, considering that the recovery periods were also longer after the longer periods of exertion.

The results may suggest that shorter sets can be beneficial to maintain external load demands without resulting in large increases in the perceived effort or heart rate responses, while ensuring a greater stimulus in total distance, running distance and number of accelerations and decelerations. However, in terms of acceleration and deceleration profiles, it may also be appropriate to choose longer sets as these contribute to a smaller decrease across sets (decrements are not so meaningful between sets as in the shorter intervals).

## Figures and Tables

**Figure 1 ijerph-16-02923-f001:**
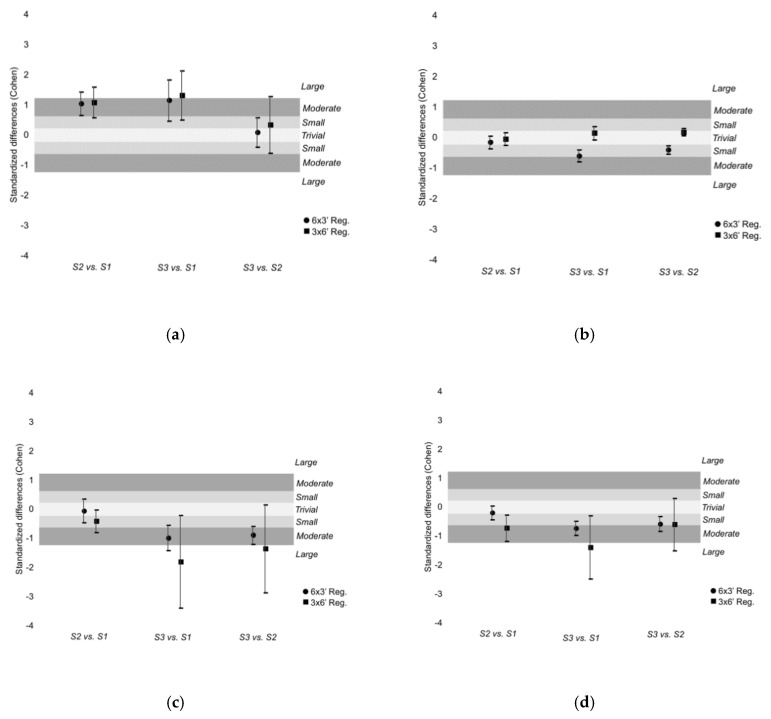
Standardized difference (Cohen) between sets in (**a**) RPE; (**b**) HRmean; (**c**) Total Distance; and (**d**) Player Load. The six sets of the 6 × 3’ regimen were grouped in three sets (S1: mean of set 1 and 2; S2: mean of set 3 and 4; S3: mean of set 5 and 6) to a better visualization and analysis. Standardized value direction depends on the relationship A-B.

**Figure 2 ijerph-16-02923-f002:**
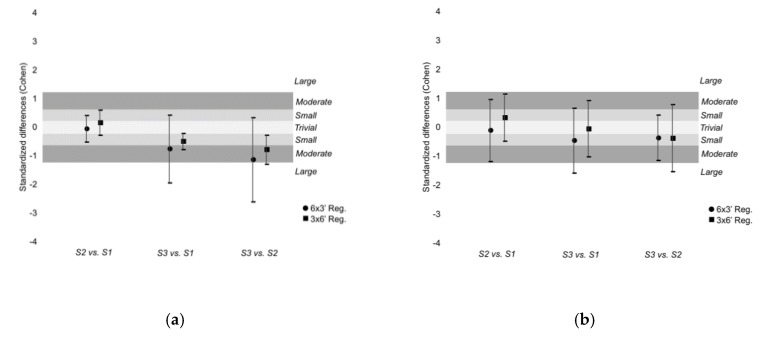
Standardized difference (Cohen) between sets in (**a**) running distance; (**b**) sprinting distance; (**c**) total accelerations; and (**d**) total decelerations. The six sets of the 6 × 3’ regimen were grouped in three sets (S1: mean of set 1 and 2; S2: mean of set 3 and 4; S3: mean of set 5 and 6) to a better visualization and analysis. Standardized value direction depends on the relationship A-B.

**Table 1 ijerph-16-02923-t001:** Descriptive statistics (Mean (Standard Deviation)) of internal and external load during the SSGs.

Variable	6 × 3’ Regimen						3 × 6’Regimen		
	S1	S2	S3	S4	S5	S6	S1	S2	S3
RPE (A.U.)	4.2 (1.5)	4.9 (1.3)	5.6 (1.4)	6.0 (1.3)	5.9 (1.3)	6.1 (1.9)	5.0 (1.2)	6.3 (1.1)	6.7 (1.6)
HRmean (bpm)	170.9 (11.1)	171.6 (10.0)	169.8 (10.9)	170.2 (12.5)	166.3 (10.8)	165.2 (12.4)	169.1 (11.7)	168.8 (10.5)	171.1 (10.9)
TD (m/min)	107.7 (10.2)	111.7 (9.1)	112.5 (11.1)	106.8 (9.8)	104.1 (8.8)	101.6 (10.9)	103.3 (7.6)	99.8 (6.9)	90.9 (15.6)
RD (m/min)	10.0 (5.1)	14.1 (5.7)	12.8(4.7)	10.6 (5.2)	11.1 (5.6)	9.1 (6.6)	8.7 (5.5)	9.4 (5.6)	6.0 (3.8)
SD (m/min)	0.5 (1.1)	1.0 (1.6)	1.5 (1.9)	0.7 (1.2)	1.0 (1.6)	0.3 (0.7)	0.6 (0.9)	0.7 (1.3)	0.5 (0.8)
TAc (n/min)	2.7 (0.9)	2.8 (0.6)	2.9 (0.8)	2.2 (1.0)	2.2 (1.0)	2.4 (1.2)	2.1 (1.0)	1.8 (0.9)	1.9 (1.1)
TDc (n/min)	2.4 (1.1)	2.5 (0.6)	2.7 (0.9)	1.7 (0.9)	2.1 (1.0)	2.1 (1.2)	1.9 (1.0)	1.7 (0.9)	1.6 (0.9)
PL (g/min)	7.2 (1.2)	7.3 (1.3)	7.3 (1.3)	6.8 (1.3)	6.4 (1.1)	6.5 (1.1)	6.9 (0.9)	6.3 (0.8)	5.9 (1.3)

SSGs: small-sided games. RPE: rated of perceived exertion (CR-10 scale); HRmean: mean heart rate; TD: total distance; RD: running distance; SD: sprinting distance; TAc: total accelerations; TDc: total decelerations; PL: player’s load; S: set; A.U.: arbitrary units; bpm: beats per minute; m/min: meters per minute; n/min: number per minute: g/min: g per minute.

**Table 2 ijerph-16-02923-t002:** Comparison of internal and external load variables between training regimens in terms of percentage and standardized differences and the probabilities of each standardized difference.

Variable	M (SD) 3 × 6’ Reg.	M (SD) 6 × 3’ Reg.	% Difference (3 × 6’ Reg.–6 × 3’ Reg.)		Standardized Difference (3 × 6’ Reg.–6 × 3’ Reg.)		% Greater/Similar/Lower Values for 3 × 6’ Reg. vs. 6 × 3’ Reg.
			Value	[90% CI]	Value (Magnitude)	90% CI	
RPE (A.U.)	5.97 (0.80)	5.43 (0.91)	10.7	[2.9; 19.2]	0.49 small	[0.14; 0.84]	92/8/0 Likely
HRmean (bpm)	169.67 (10.05)	169.52 (9.75)	0.1	[−1.8; 2.0]	0.01 trivial	[−0.28; 0.31]	14/75/11 Unclear
TD (m/min)	98.39 (7.49)	107.56 (5.99)	−8.6	[−12.3; −4.8]	−1.51 large	[−2.20; −0.82]	0/0/100 Almost certain
RD (m/min)	8.04 (3.31)	11.28 (1.87)	−34.0	[−47.0; −17.8]	−2.23 large	[−3.40; −1.05]	0/0/100 Almost certain
SD (m/min)	0.62 (0.42)	0.83 (0.49)	−13.9	[−46.7; 39.3]	−0.13 trivial	[−0.54; 0.29]	9/53/38 Unclear
TAc (n/min)	1.95 (0.55)	2.53 (0.54)	−24.0	[−35.1; −10.9]	−1.11 moderate	[−1.75; −0.47]	0/1/99 Very likely
TDc (n/min)	1.70 (0.53)	2.24 (0.39)	−26.7	[−38.8; −12.1]	−1.49 large	[−2.36; −0.62]	0/1/99 Very likely
PL (g/min)	6.37 (0.79)	6.92 (0.99)	−7.8	[−12.6; −2.7]	−0.58 small	[−0.97; −0.20]	0/5/95 Likely

RPE: rated of perceived exertion (CR-10 scale); HRmean: mean heart rate; TD: total distance; RD: running distance; SD: sprinting distance; TAc: total accelerations (>2 m/s^2^); TDc: total decelerations (>2 m/s^2^); PL: g/min; S: set; Reg.: training regimen; A.U.: arbitrary units; bpm: beats per minute; m/min: meters per minute; n/min: number per minute: g/min: g per minute.

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
