# Peer review of "Variations of Internal and External Load Variables between Intermittent Small-Sided Soccer Game Training Regimens"

_ijerph, 2019, doi:10.3390/ijerph16162923_

Round 1
Reviewer 1 Report
I have to admit that it is a technically (method) very serious work, Congratulations for the good job done.
Lets see a few concerns or doubts i had:
Abstract
-Please clarify RPE in the Abstract.
Introduction
-Professor Castellano must appear with at least three or four newest papers in introduction and discussion. Please do this search job: Julen Castellano (University of Pais Vasco, Spain).
- Please write an introductory paragraph to address the complexity of the decision when players playing (Aguilar et al 2018; Pic, et al 2018). How it can affect (decisional complexity) to soccer players in specific situations 5x4 (Méndez-Dominguez, Gómez Ruano et al 2019) for instance with intermittent regimens (6x3’ and 3x6’)?
References to be used:
Aguilar, M.; Navarro-Adelantado, V., & Jonsson, G. K. (2018). Detection of Ludic Patterns in Two Triadic Motor Games and Differences in Decision Complexity. Frontiers in Psychology, 2259(8), 1-10. https://doi.org/10.3389/fpsyg.2017.02259
Pic, M., Lavega-Burgués, P., & March-Llanes, J. (2018). Motor behaviour through traditional games. Educational Studies, 1-14.
Méndez-Domínguez, C., Gómez-Ruano, M. A., Rúiz-Pérez, L. M., & Travassos, B. (2019). Goals scored and received in 5vs4 GK game strategy are constrained by critical moment and situational variables in elite futsal. Journal of sports sciences, 1-9.
Method
Materials and Methods2.1. Subjects
-‘2.1. Subjects’ should be changed by ‘participants’
-Is any university involved in the study? Please insert the name.
2.2. Experimental Approach
- Please, why should be used 5x5 format? The size of the 105 pitch was 42x22 suitable with soccer? Please, write about the importance of the concept of players interations (Parlebas, 2001) (with partner and rival).
2.3. Small-sided game
- Why are there no goalkeepers in the goal? This could make the transfer to practice difficult.
2.6. Statistical Procedures
- Please should be insert information related to the data normality
- Why were used so low 90% confidence intervals (90% CI) instead of higher? I expect you to explain or/and being included in limitations of the study.
Author Response
REVIEWER 1
I have to admit that it is a technically (method) very serious work, Congratulations for the good job done.
AUTHORS: DEAR REVIEWER, THANK YOU SO MUCH FOR YOUR COMMENTS AND RECOMMENDATIONS. WE HAVE FOLLOWED ALL YOUR SUGGESTION TO IMPROVE THE ARTICLE.
Lets see a few concerns or doubts i had:
Abstract
-Please clarify RPE in the Abstract.
AUTHORS: DEAR REVIEWER, THANK YOU. WE HAVE CHANGED ACCORDINGLY.
Introduction
-Professor Castellano must appear with at least three or four newest papers in introduction and discussion. Please do this search job: Julen Castellano (University of Pais Vasco, Spain).
AUTHORS: DEAR REVIEWER, THANK YOU. WE HAVE ADDED NEW REFERENCES. JULEN APPEARS NOW IN THREE REFERENCES IN INTRODUCTION AND DISCUSSION.
- Please write an introductory paragraph to address the complexity of the decision when players playing (Aguilar et al 2018; Pic, et al 2018). How it can affect (decisional complexity) to soccer players in specific situations 5x4 (Méndez-Dominguez, Gómez Ruano et al 2019) for instance with intermittent regimens (6x3’ and 3x6’)?
References to be used:
Aguilar, M.; Navarro-Adelantado, V., & Jonsson, G. K. (2018). Detection of Ludic Patterns in Two Triadic Motor Games and Differences in Decision Complexity. Frontiers in Psychology, 2259(8), 1-10. https://doi.org/10.3389/fpsyg.2017.02259
Pic, M., Lavega-Burgués, P., & March-Llanes, J. (2018). Motor behaviour through traditional games. Educational Studies, 1-14.
Méndez-Domínguez, C., Gómez-Ruano, M. A., Rúiz-Pérez, L. M., & Travassos, B. (2019). Goals scored and received in 5vs4 GK game strategy are constrained by critical moment and situational variables in elite futsal. Journal of sports sciences, 1-9.
AUTHORS: DEAR REVIEWER, THANK YOU. WE HAVE FOLLOWED YOUR RECOMMENDATION. A NEW PARAGRAPH WAS WRITEN BASED ON THE REFERENCES THAT YOU SUGGESTED.
Method
Materials and Methods
2.1. Subjects
-‘2.1. Subjects’ should be changed by ‘participants’
AUTHORS: DEAR REVIEWER, THANK YOU. WE HAVE CHANGED ACCORDINGLY.
-Is any university involved in the study? Please insert the name.
AUTHORS: DEAR REVIEWER, THANK YOU. WE HAVE ADDED THE NAME OF THE UNIVERSITY.
2.2. Experimental Approach
- Please, why should be used 5x5 format? The size of the 105 pitch was 42x22 suitable with soccer? Please, write about the importance of the concept of players interations (Parlebas, 2001) (with partner and rival).
AUTHORS: DEAR REVIEWER, THANK YOU. WE HAVE ADDED THE JUSTIFICATION BASED ON A STUDY THAT ANALYSED THE AREA OF PLAY DURING OFFICIAL MATCHES AND PROPOSED THE AREA FOR SSGs.
2.3. Small-sided game
- Why are there no goalkeepers in the goal? This could make the transfer to practice difficult.
AUTHORS: DEAR REVIEWER, THANK YOU. WE HAVE USED WITHOUT GK TO ELICT A GREAT INTENSITY THAT IS USUALLY FOUND IN SSGs WITHOUT GK.
2.6. Statistical Procedures
- Please should be insert information related to the data normality
AUTHORS: DEAR REVIEWER, THANK YOU. WE HAVE ADDED THE INFORMATION.
- Why were used so low 90% confidence intervals (90% CI) instead of higher? I expect you to explain or/and being included in limitations of the study.
AUTHORS: DEAR REVIEWER, THANK YOU. WE HAVE ADDED AN EXPLANATION IN-TEXT.
Reviewer 2 Report
Ladies and gentlemen,
based on the article sent for review, I find that:
the title of the work is consistent with the content presented in it the first of my remarks concerns the legitimacy of presenting very detailed research results in the abstract. The abstract should discuss the presented assumptions, results and conclusions, and at the same time encourage to get acquainted with the whole material the introduction adequately determines the validity of the decision to conduct the tests description of the test method (indicators and methods of measuring them) is correct. However, the question concerns the place where the research was carried out, including external conditions - this information was not provided the variables have been described at a satisfactory level the statistical analyzes used are well matched when it comes to verification of assumed research hypotheses the data presented in the tables and charts are clear and clearly show the results obtained legitimate content regarding limitations occurring in the research (the external conditions prevailing during the experiment were described to a limited extent) the final size of the research group is not clearly specified in the article the description of the results is in many places a repetition of the data contained in the tables the literature cited to the article is chosen correctly I believe that the research and the results obtained contribute, in my opinion, to relevant knowledge useful for the development of the studied area of ​​sports scienceAuthor Response
REVIEWER 2
Ladies and gentlemen,
based on the article sent for review, I find that:
the title of the work is consistent with the content presented in it the first of my remarks concerns the legitimacy of presenting very detailed research results in the abstract. The abstract should discuss the presented assumptions, results and conclusions, and at the same time encourage to get acquainted with the whole material the introduction adequately determines the validity of the decision to conduct the tests description of the test method (indicators and methods of measuring them) is correct. However, the question concerns the place where the research was carried out, including external conditions - this information was not provided the variables have been described at a satisfactory level the statistical analyzes used are well matched when it comes to verification of assumed research hypotheses the data presented in the tables and charts are clear and clearly show the results obtained legitimate content regarding limitations occurring in the research (the external conditions prevailing during the experiment were described to a limited extent) the final size of the research group is not clearly specified in the article the description of the results is in many places a repetition of the data contained in the tables the literature cited to the article is chosen correctly I believe that the research and the results obtained contribute, in my opinion, to relevant knowledge useful for the development of the studied area of ​​sports science
AUTHORS: DEAR REVIEWER, THANK YOU SO MUCH FOR YOUR COMMENTS. WE HAVE TRIED TO IMPROVE THE MANUSCRIPT STRICLY FOLLOWING YOUR SUGGESTIONS.
Reviewer 3 Report
This manuscript reports on an experimental study in which the effects of two different intermittent small-sided soccer game training regimens on physiological and psychological variables were examined. It is not immediately apparent that the topic addressed in the manuscript falls within the domain covered by this journal (according to the “Aims & Scope”). The manuscript is generally well-written. These impressions notwithstanding, I have several concerns about the manuscript in its current form:
On l. 68, it should be “durations.”
With respect to the text on l. 86, “cross-sectional” should be replaced another term that more accurately describes the design. The design is not cross-sectional; it is a repeated-measures experimental design with order counterbalanced.
To what extent did urging participants “to maintain a high exertion level” (l. 110-111) bias participants’ ratings of perceived exertion?
On l. 113, it should be “Rating of Perceived Exertion (RPE).”
Why weren’t repeated-measures ANOVAs (or MANOVAs) conducted along with an assessment of the effect of condition presentation order?
On l. 267, it should be “led.”
I appreciate the authors’ mention of work-to-rest ratios in the paragraph on limitations of the study (l. 253-266). This issue deserves more attention. In the current study, the two regimens differed in terms of number of sets, duration of sets, and amount of rest (4 min v 10 min) or work-rest ratio (18:4 v 18:10). The total duration of work was constant across the two regimens. Observed differences in the outcome variables cannot be attributed to any one of the characteristics of the regimens in isolation. Consequently, any differences between the two regimens have to be attributed to the entire combination of distinguishing features of the regimens (number of sets, duration of sets, and amount of rest/work-rest ratio). Instead, conclusions in this study seem to be geared around the “longer sets” versus “shorter sets” distinction.
To interpret the results of the study, it is necessary to know which of the two regimens generalizes most closely to the task demands of full-sided play and, especially, which best enhances performance in full-sided play. Have any small-sided regimens been tested or compared along these criteria?
Author Response
Reviewer 3
This manuscript reports on an experimental study in which the effects of two different intermittent small-sided soccer game training regimens on physiological and psychological variables were examined. It is not immediately apparent that the topic addressed in the manuscript falls within the domain covered by this journal (according to the “Aims & Scope”). The manuscript is generally well-written. These impressions notwithstanding, I have several concerns about the manuscript in its current form:
AUTHORS: DEAR REVIEWER, THANK YOU SO MUCH FOR YOUR COMMENTS AND SUGGESTIONS. WE DO BELIEVE THAT FIT IN THE SCOPE OF THE SPECIAL ISSUE “Health, Exercise and Sports Performance”, namely in the sports performance.
On l. 68, it should be “durations.”
AUTHORS: DEAR REVIEWER, THANK YOU. WE HAVE CHANGED ACCORDINGLY.
With respect to the text on l. 86, “cross-sectional” should be replaced another term that more accurately describes the design. The design is not cross-sectional; it is a repeated-measures experimental design with order counterbalanced.
AUTHORS: DEAR REVIEWER, THANK YOU. WE AGREE WITH YOU. WE HAVE CHANGED ACCORDINGLY.
To what extent did urging participants “to maintain a high exertion level” (l. 110-111) bias participants’ ratings of perceived exertion?
AUTHORS: DEAR REVIEWER, THANK YOU. WE HAVE USED A PREVIOUS STUDY OF RAMPININI ET AL TO APPLY SIMPLE VERBAL ENCOURAGMENT TO ENSURE THE HIGHER EXERTION AS POSSIBLE. THIS MAY HELP IN TO MAXIMIZE THE EFFORT IN EVERY BOUT.
AUTHORS:
On l. 113, it should be “Rating of Perceived Exertion (RPE).”
AUTHORS: DEAR REVIEWER, THANK YOU. WE HAVE CHANGED ACCORDINGLY.
Why weren’t repeated-measures ANOVAs (or MANOVAs) conducted along with an assessment of the effect of condition presentation order?
AUTHORS: DEAR REVIEWER, THANK YOU. WE HAVE USED A MAGNITUDE-BASED EVIDENCE APPROACH TO THE STATISTICAL ANALYSIS.
On l. 267, it should be “led.”
AUTHORS: DEAR REVIEWER, THANK YOU. WE HAVE CHANGED ACCORDINGLY.
I appreciate the authors’ mention of work-to-rest ratios in the paragraph on limitations of the study (l. 253-266). This issue deserves more attention. In the current study, the two regimens differed in terms of number of sets, duration of sets, and amount of rest (4 min v 10 min) or work-rest ratio (18:4 v 18:10). The total duration of work was constant across the two regimens. Observed differences in the outcome variables cannot be attributed to any one of the characteristics of the regimens in isolation. Consequently, any differences between the two regimens have to be attributed to the entire combination of distinguishing features of the regimens (number of sets, duration of sets, and amount of rest/work-rest ratio). Instead, conclusions in this study seem to be geared around the “longer sets” versus “shorter sets” distinction.
AUTHORS: DEAR REVIEWER, THANK YOU. WE DO AGREE WITH YOU. THUS, WE HAVE EXTENDED THE STUDY LIMITATIONS SECTION AND WE HAVE ADDED A NEW SENTENCE IN THE CONCLUSIONS ABOUT SUCH FACT.
To interpret the results of the study, it is necessary to know which of the two regimens generalizes most closely to the task demands of full-sided play and, especially, which best enhances performance in full-sided play. Have any small-sided regimens been tested or compared along these criteria?
AUTHORS: DEAR AUTHOR, THANK YOU FOR THE COMMENT. WE HAVE EXTENDED THE PRATICAL IMPLICATIONS BASED ON YOUR SUGGESTION.